# Early T-Cell Precursor ALL and Beyond: Immature and Ambiguous Lineage T-ALL Subsets

**DOI:** 10.3390/cancers14081873

**Published:** 2022-04-08

**Authors:** Eulàlia Genescà, Roberta la Starza

**Affiliations:** 1Institut d’Investigació Contra la Leucemia Josep Carreras (IJC), Campus ICO-Germans Trias i Pujol, Universitat Autònoma de Barcelona, 08916 Badalona, Spain; 2Hematology and Immunology Section, Department of Medicine and Surgery, CREO, Università degli Studi di Perugia, 06132 Perugia, Italy

**Keywords:** immature T-ALL, ETP-ALL, diagnosis, genomics, outcome, treatment

## Abstract

**Simple Summary:**

Immature T-cell acute lymphoblastic leukemias englobes a wide range of low prevalence subtypes, not well identified, that in some cases overlap with myeloid lineage subtypes. Globally, this “grey zone” of immature leukemias, are difficult to precisely diagnose using a classical immunophenotypic approach. Interesting, genomic data collected during last years has shown that these subtypes share several genomic alterations, raising the question of how their phenotypes reflect distinct AL entities. Here we provide a systematic overview of the genetic events associated with immature T-ALL and outline their relationship with treatment choices and outcomes. Our goal is to offer a basis for using the genetic information for new diagnostic algorithms. An immunogenetic classification of these immature subtypes will better stratify patients and improve their management with more efficient and personalized therapeutic options.

**Abstract:**

A wide range of immature acute leukemias (AL), ranging from acute myeloid leukemias with minimal differentiation to acute leukemias with an ambiguous lineage, i.e., acute undifferentiated leukemias and mixed phenotype acute leukemia with T- or B-plus myeloid markers, cannot be definitely assigned to a single cell lineage. This somewhat “grey zone” of AL expresses partly overlapping features with the most immature forms of T-cell acute lymphoblastic leukemia (T-ALL), i.e., early T-cell precursor ALL (ETP-ALL), near-ETP-ALL, and pro-T ALL. These are troublesome cases in terms of precise diagnosis because of their similarities and overlapping phenotypic features. Moreover, it has become evident that they share several genomic alterations, raising the question of how their phenotypes reflect distinct AL entities. The aim of this review was to provide a systematic overview of the genetic events associated with immature T-ALL and outline their relationship with treatment choices and outcomes, especially looking at the most recent preclinical and clinical studies. We wish to offer a basis for using the genetic information for new diagnostic algorithms, in order to better stratify patients and improve their management with more efficient and personalized therapeutic options. Understanding the genetic profile of this high-risk T-ALL subset is a prerequisite for changing the current clinical scenario.

## 1. Introduction

### 1.1. The Grey Zone of Immature Leukemias

Although several recurrent genetic events are among the diagnostic hallmarks of specific leukemia subtypes, an immediate AL diagnosis relies on morphological and immunophenotypic features. Once the morphology has been assessed for the presence of ≥20% marrow leukemic cells, the expression of cytoplasmic and/or surface antigens is necessary for classifying the AL as myeloid, B-, or T-lymphoid. However, besides cases responding to strict immunophenotypic criteria, there is a “grey zone” of AL including a wide range of immature ALs which either do not express clear lineage-specific antigens or, on the contrary, display a variable combination of myeloid and lymphoid antigens. This is the case with acute undifferentiated leukemias (AUL), and B/myeloid and T/Myeloid mixed phenotype acute leukemias (B/M MPAL and T/M MPAL) [1,2,3]. In this setting, we can also include AML with minimal differentiation, which is diagnosed when blasts are positive for at least two myeloid-associated antigens (usually CD117 and CD13), do not express MPO at cytochemistry (even though some blasts might be positive at flow cytometry and immunohistochemistry), and might express CD7 and CD2 antigens (roughly 40% of cases) [4].

Some of these entities have close/overlapping characteristics with the most immature forms of T-cell acute lymphoblastic leukemia (T-ALL) [2,5]. The latter are further divided into early T-cell precursor ALL (ETP-ALL), near-ETP-ALL, and pro-T ALL (Figure 1).

The tight association between T/M MPAL and immature T-ALL has become evident in the last few years, since next-generation sequencing (NGS) studies have shown a wide range of overlapping genomic events, as well as similar gene expression signatures [6]. Furthermore, several rearrangements, such as *MLLT10*, *NUP214*, and *NUP98* translocations, are also shared between T-ALL and AML M0 subtypes, suggesting that specific genetic entities possibly prevail over phenotypic clustering [7,8,9,10,11].

### 1.2. Immature T-Cell Acute Lymphoblastic Leukemia (T-ALL) 

T-ALL is a rare subtype accounting for 15% of childhood and 25% of all adult ALLs [11]. Widely considered a poor prognostic subgroup in the past, its management has greatly improved through the use of pediatric-inspired protocols in adults [12,13], including a refined risk stratification with measurable residual disease (MRD) monitoring. In several clinical trials, the overall survival was more than 80% in children and adolescents [14] and 70% in younger adults [15]. However, it is still less than 50% in older adults [13,16], and refractory/relapsed cases (R/R) do not benefit from available salvage treatments and fare very badly, either with chemotherapy regimens, such as FLAG-IDA, or with the most recently approved drug, nelarabine [17]. Both salvage schemes resulted in a response rate lower than 40% [18,19]. Therefore, therapy for R/R T-ALL represents a highly challenging unmet need, especially in adults. In this scenario, the identification of new predictive and prognostic markers would be extremely useful for fine-tuning the risk stratification and developing more effective personalized therapies. These markers should include the disease immunophenotypes and genetic alterations.

According to the immunophenotype, T-ALL is classified into four main subtypes: mature (sCD3 positive), cortical (CD1a+), pre-T (cCD3+ plus CD2+ and/or CD5+, and/or CD8+), and pro-T ALL (cCD3+ and CD7+) [2,20]. In 2009, the immature pro-T ALL subtype was more strictly defined as ETP-ALL and, later on, the so-called near-ETP (see below) was placed between ETP and early (or pre-T) ALL. Therefore, “immature T-ALL” refers to cases with slightly different patterns of antigen expression, but it is unclear whether these reflect true distinct clinical and prognostic subsets. These aspects and the lack of an in-depth characterization make immature T-ALL a high-risk subgroup both in children and adults [16,21,22,23,24,25], an interesting area of research. Moreover, dissecting the genetic background of immature T-ALL will inform about the specific leukemogenic pathways of distinctive genetic subtypes, contributing to refining the immunophenotypic classification and resolving misclassifications.

The main aim of this review was to focus on the genetic events associated with immature T-ALL and outline their relationship with treatments and outcomes. The new genetic information can be used for accurate diagnostic processes, risk stratification, and the choice of personalized and more efficient therapeutic options. 

## 2. ETP-ALL

### 2.1. Historical Perspective of Immune Markers and Gene Expression Profiles (GEPs) 

The immunological definition of ETP-ALL strictly refers to the Coustan-Smith criteria [21]. On the other hand, RNA microarray studies have also distinguished an immature cluster of T-ALLs, which are characterized by a specific genetic signature, marked by high levels of expression of *LYL1*, *LMO2*, *CD34*, *KIT*, and myeloid markers [26,27]. In 2014, a study assessed the overlap between ETP-ALL, the immature T-ALL cluster, and the so-called ABD-T-ALL, another immature T-ALL entity defined on the basis of the absence of TCRΥ biallelic deletions [28]. In comparisons of gene expression profiles, ETP-ALL and the immature T-ALL cluster emerged as a single entity in which ABD-positive cases represented a subgroup [29]. However, the original ETP-ALL classification of Coustan-Smith and co-authors significantly underestimates the number of actual patients with an immature T-ALL profile [29], revealing discrepancies between the ETP subgroups defined by GEP or by immunophenotype.

During the last few years, integrated genetic approaches have provided new insights into the complex and heterogeneous group of immature T-ALL/ETP-ALL (see below), and whole-genome sequencing (WGS) has shown that the mutational spectrum of immature T-ALL has close similarities to AML, while RNA sequencing has revealed that the transcriptional profile of immature T-ALL resembles that of normal hematopoietic stem cells and AML [30]. Until now, the new genetic information has not been translated into the clinics, and we are still missing a reliable diagnostic algorithm to precisely identify all T-ALL cases that belong to the distinct immature subtypes and assess the clinical impact this may exert on the response to therapy and outcome.

### 2.2. Diagnosis

The application of large panels of monoclonal antibodies in flow cytometry is currently the diagnostic test for distinguishing among AML with minimal differentiation (herein M0, according to the original FAB classification), AUL, T/M MPAL, ETP-ALL, near-ETP-ALL, and pro-T ALL. Strictly following the WHO criteria, AML-M0 is characterized by the expression of stem cell (CD117, CD38) and/or myeloid (HLA-DR, CD33, CD13) markers without any other lineage-specific antigen; AUL is defined by the absence of any lineage-specific antigen, i.e., MPO, cCD3, CD79a, and cCD22. The diagnosis of T/M MPAL instead requires the expression of both the myeloid marker MPO or at least two monocytic markers (NSE, CD11c, CD14, CD64, lysozyme) and cytoplasmic (or more rarely surface) CD3, often co-expressed with additional myeloid and T-lymphoid markers (Table 1).

A diagnosis of ETP-ALL is based on the expression of cytoplasmic CD3, CD7, and CD2, with a lack of CD1a and CD8, and positivity for one or more of the myeloid/stem cell markers CD34, CD117, HLADR, CD13, CD33, CD11b, or CD65 [5,21]. Blasts may express CD2 and CD4, but these antigens are not part of the definition. The CD5 marker must be negative or, if positive, it must be expressed by less than 75% of the blast cell population (Table 2) [5,21]. However, the “partial CD5 expression” criterion has a negative impact on the reproducibility of ETP-ALL diagnosis because of a lack of standardization for the method. Zuubier et al. proposed refined immunophenotypic criteria by excluding CD5 expression while adding negativity for CD4 [29]. Actually, there are cases with an immature phenotype which, however, are positive for CD5. These cases have been termed “near-ETP-ALL” and are kept separate from both ETP and typical T-ALL (Table 2) [11,31,32]. Whether ETP and near-ETP-ALL represent two distinct entities with unique clinical/hematological characteristics remains to be elucidated.

### 2.3. Genomic Characterization 

A compendium of comprehensive molecular-cytogenetic studies and sequencing approaches that are used to characterize this rare subtype of leukemia have provided an overall view of the genes, pathways, and events associated with ETP-ALL that are often shared with other immature ALs, mainly T/M MPAL [6,10,31].

Molecular-cytogenetic studies and sequencing have shown that alterations that are highly recurrent in typical T-ALL do not occur or are seldom detectable in immature T-ALL. Examples are mutations of T-ALL-related oncogenes/onco-suppressors, such as *NOTCH1*, *FBXW7*, *PTEN*, and *RPL10*; deletions of *CDKN2AB*; and translocations in T-related transcription factors, such as *TAL1*, *LMO1*, *LMO2*, *MYB*, *NKX2.1*, and *TLX1* [11,31,33,34,35]. On the other hand, WGS has shown that 81% of pediatric ETP-ALL cases harbor mutations in genes regulating hematopoietic development (i.e., *GATA3*, *RUNX1*, *ETV6*, *IKZF1*, and *EP300*), in cytokine receptors and members of the RAS signaling pathway (i.e., *BRAF*, *FLT3*, *IGFR1*, *JAK1*, *JAK3*, *KRAS*, and *NRAS*), and in components of the polycomb repressor complex 2 (PRC2) (i.e., *EED*, *EZH2*, and/or *SUZ12*) [30]. The same analysis performed in adult ETP-ALL cases revealed specific genetic alterations that differ from those found in childhood ETP-ALL. For example, *DNMT3A* mutations are typical of adulthood [16,36,37], being more frequent in patients aged >60 years (12%) [38]. Other genetic alterations prevailing in adult ETP-ALL are the FLT3 mutations. In fact, either an internal tandem duplication (FLT3-ITD) or point mutations in the tyrosine kinase domain (FLT3-TKD) [39,40] have been detected in 35% of adult and less than 10% of pediatric ETP-ALL cases [30,41].

Much less is known regarding structural variants. A significantly higher rate of copy number alterations (CNA), including both genomic gains and losses, have been reported in pediatric ETP-ALL, and the overall size of the genomic regions undergoing CNA are significantly larger in ETP-ALL than in non-ETP cases (*p* = 0.0068) [21]. Interestingly, among gene deletions, 13q losses are significantly associated with an ETP phenotype (*p* = 0.0095) [21]. In contrast, deletions of *CDKN2A/B* are detected in less than 30% of ETP-ALL [33,34,35].

Pioneering RNA microarray studies first demonstrated that immature T-ALLs are characterized by higher levels of *LMO1*, *LYL1*, and *ERG* expression than non-immature cases [26]. More recently, *MEF2C* transcription factor (TF) was also observed to be overexpressed in immature T-ALLs [42].

In vitro functional experiments in cell lines indicated that *MEF2C* functions as a transcriptional regulator, resulting in the activation of at least five genes highly expressed in immature T-ALL, namely *PSCD4*, *HHEX*, *FAM46A*, *LMO2*, and *LYL1*. Consistent with this finding, *MEF2C* knockdown led to reduced levels of *LMO2*, *HHEX*, and *LYL1*, suggesting that the activation of these genes, through *MEF2C*, may be crucial for early T cell engagement [41]. Although it is known that they are highly expressed in hematopoietic tissue (*HHEX* and *FAM46A*) or peripheral blood (*PSCD4*), to date, no further information is available on the possible interactions and reciprocal inter-relationships between *MEF2C* and *HHEX*, *PSCD4*, or *FAM46A* (https://www.genecards.org, accessed on 30 March 2022).

A global comparison of the gene expression profile of ETP-ALL with that obtained from purified normal hematopoietic stem cells, progenitor cell populations, and myeloid blast cells has shown an enrichment in normal hematopoietic stem cell- and granulocyte-macrophage precursor cell-expressed genes in ETP-ALL, but not in genes expressed in normal human early T-cell precursors [29]. Further gene expression analysis of ETP-ALL has revealed the participation of non-coding RNA in the development of this leukemia subtype. A significant upregulation of miR-221 and miR-222, two putative oncogenes, has been observed in adult ETP-ALL [43]. Importantly, the miR-222 target ETS1 was specifically downregulated in ETP-ALL cases, and in vitro experiments in transfected T-ALL cell lines showed that it inhibited cell proliferation and induced apoptosis (Figure 2) [43].

Taken together, genetic and expression studies indicate that genes altered in ETP-ALL are more likely to be affected in AML [44] than in non-immature T-ALL. These findings suggest that the initial genetic lesions occur in a poorly differentiated stem-cell-like/myeloid-lymphoid precursor. To date, the wide heterogeneity and complexity of ETP-ALL, together with the rarity of the disease, has not allowed us to recognize distinct genetic entities characterized by unique genomic and transcriptomic profiles, in spite of several attempts. Nonetheless, we wish to mention two relevant genetic subgroups defined by primary events.

#### 2.3.1. Genomic Alterations That Cause HOXA Overexpression

Although genomic abnormalities that cause an overexpression of *HOXA* genes can be detected in different subtypes of T-ALL, a close association with an immature phenotype has been reported [45,46,47,48]. Actually, abnormalities that cause *HOXA* upregulation, can be detected in 30–40% of ETP/near-ETP cases [46,48]. In fact, *HOXA* genes are *cis*-activated by genomic rearrangements with *TR@* as well as no-*TR@* loci (Table 3) [49,50], but they can also be *trans*-activated by rearrangements affecting *KMT2A*, *MLLT10*, *NUP214*, or *NUP98*. Although they occur in both typical and immature T-ALL cases, *trans*-activation is highly recurrent in the latter [46]. Due to the heterogeneity of the genomic background and the promiscuous behavior of the main T-ALL oncogenes, *HOXA*-positive cases are possibly underestimated [48]. Here, we provide a summary (Table 3).

*PICALM-MLLT10*, the most frequent fusion transcript in T-ALL, accounts for 7% of pediatric and 6% of adult T-ALL cases [51]. This fusion, manly resulting from t(10;11)(p12;q14), identifies a poor prognostic subgroup within sCD3/T-cell receptor-negative adult T-ALL. Among adult ETP-ALL cases, *PICALM-MLLT10* identifies patients who are likely to fail to obtain remission [51]. In a similar way, rearrangements affecting *KMT2A* (*MLL*), including *KMT2A-MLLT4, KMT2A-MLLT1, KMT2A-ELN, KMT2A-ELL*, and *KMT2A-AFDN*, have been associated with ETP-ALL phenotypes that were more likely to fail therapy, especially when *KMT2A-AFDN* was involved [52]. Lastly, *SET-NUP214* and *SQSTM1-NUP214* fusions, although present in less than 5% of cases, appear to be associated with a dismal outcome [53]. It is worth remembering that all of these oncogenes can also be involved in AML, most commonly immature and undifferentiated AL.

**Table 3 cancers-14-01873-t003:** Known rearrangements that *cis*- or *trans*-activate the *HOXA* gene cluster in immature T-ALL.

*HOXA* [10,30,47]	*KMT2A* [46,52]	*MLLT10* [30,46,52]	*NUP214* [10,46,52]	*NUP98* [46]
*mir181::HOXA*	*KMT2A::MLLT4*	*PICALM::MLLT10*	*SET::NUP214*	*NUP98::RAP1GDS1*
*TRB@::-HOXA*	*KMT2A::AFDN*	*DDX3X::MLLT10*	*SQSTM1::NUP214*	*NUP98::PSIP1*
*CDK6::HOXA*	*KMT2A::ELL*	*HNRNPH1::MLLT10*		*NUP98::DDX10*
*TRG@::HOXA*	*KMT2A::ENL*	*NAP1L1::-MLLT10*		*NUP98::VRK1*
*TRAD@::HOXA*	*KMT2A::CBL*	*CAPS2::MLLT10*		*NUP98::LNP1*
	*KMT2A::MLLT1*	*XPO1::MLLT10*		*NUP98::CCDC28A*
	*KMT2A::MLLT10*			
	*KMT2A::CT45A4*			
	*KMT2A::MLLT6*			

#### 2.3.2. The *BCL11B*-a Entity

A unique genetic subtype has been recently identified among immature ALs, ranging from ETP-ALL to immature AML [10,54]. This entity is marked by rearrangements involving the long arm of chromosome 14 at 14q32 that consist of translocations/insertions, with different partners, or focal tandem duplications that generate a super-enhancer in the non-coding sequences placed at 750 kb downstream from *BCL11B* (*BCL11B* enhancer tandem amplification or BETA). Because all these alterations invariably cause the transcriptional activation of the *BCL11B* gene, this leukemia subtype was called “*BCL11B* activated” AL (*BCL11B*-a AL). To date, four types of translocation have been identified, i.e., t(2;14)(q22.3;q32), t(6;14)(q25.3;q32), t(7;14)(q21.2;q32), and t(8;14)(q24.2;q32), which, although they are cryptic in conventional cytogenetics in some cases, can be reliably identified with specific fluorescence in situ hybridization (FISH) assays. However, diagnostic tools to pick up the focal duplication that generates the *BCL11B* super-enhancer are not yet available [10], and we do not know whether *BCL11B* rearrangements with *SATB1*, *ETV6*, and *RUNX1,* identified by RNA sequencing, can be detected by molecular-cytogenetic tools [54]. Table 4 summarize genetic alterations leading on to *BCL11B* expression activation.

*BCL11B*-a AL have a distinct expression profile, characterized by deregulation of *BCL11B*’s targets, inhibition of the T-cell differentiation program, and activation of the JAK/STAT transduction pathway. Interestingly, as predicted by the genomic profile, *BCL11B*-a AL cases appear to be extremely sensitive to tyrosine kinase and JAK/STAT inhibitors in ex vivo experiments [10,54]. Some new data that strengthen the leukemogenic role of BCL11B in immature AL came from the identification of a circular RNA (circRNA) derived from *BCL11B* expression. The so-called circ*BCL11B* has been exclusively detected in AML with a T-cell-like gene expression signature [55]. Interestingly, knockdown of this circ*BCL11B* had a negative effect on leukemic cell proliferation and resulted in the increased cell death of leukemic cells [55]. As the molecular-cytogenetic data were not available for the circ*BCL11B*-positive cases, it was not possible to link the expression of this circulating RNA with the presence of specific genetic alterations of *BCL11B*.

### 2.4. Outcomes

An early study based on limited numbers of pediatric patients indicated that ETP-ALL was associated with a very poor outcome (St. Jude Hospital trials XIII, XIV, and XV) [56]. In this study, which included 139 T-ALL patients with 12.6% diagnosed with ETP-ALL, the cases with an immature phenotype had a significantly worse outcome (*p* < 0.0001 for OS and EFS), with a 10-year OS of 19% (95% CI: 0–92). Moreover, the incidence of hematological relapses (RE) was significantly higher in the ETP-ALL subgroup (*p* < 0.0001), with a 10-year RE of 72% (95% CI: 40–100). Similar results were obtained in three additional pediatric series: the first included 100 patients, with three ETP-ALL cases identified, who were enrolled in the ALL-2000 protocol of the AIEOP (Associazione Italiana Ematologia Oncologia Pediatrica) [20]; the second had 91 patients enrolled in the Tokyo Children’s Cancer Study Group L99-15 [21], of whom five were diagnosed as ETP-ALL; and the last comprised 74 patients, 12 of whom had ETP-ALL, from the First Affiliated Hospital, College of Medicine, Zhejiang University [23].

Data obtained from larger series of children, treated with more intensive therapy, have not confirmed these differences. The Medical Research Council UKALL 2003 trial [57], conducted on 222 cases with 16% diagnosed with ETP-ALL, did not find any difference in the 5-year EFS (ETP 76.7% vs. 84.6% in non-ETP, *p* = 0.2) and OS (ETP 82.4% vs. 90.9% in non-ETP, *p* = 0.1) or in the relapse rate (ETP 18.6% vs. 9.6% in non-ETP, *p* = 0.1) [57]. Similarly, the Children’s Oncology Group (COG) AALL0434 treatment study [14] showed that the outcome was not significantly worse in patients with ETP ALL (5-year EFS and OS rates for ETP of 87.0% and 93.0%, and 86.9% and 92.0% for non-ETP ALL, respectively), although they experienced a higher induction failure rate (7.8% vs. 1.1% with an MRD higher than 0.01% by flow cytometry) at the end of the induction treatment in most of the ETP-ALL cases [14]. Together, these studies show that in the setting of pediatric ALL, MRD-based treatment regimens are able to overcome the unfavorable prognosis of high-risk cases, which, once identified, can be subjected to intensified treatments with or without hematopoietic stem cell transplantation.

Unlike pediatric cases, limited data and conflicting results have been reported for adult ETP-ALL patients. In the MD Anderson study, ETP-ALL patients (17% of the entire cohort) experienced complete remission (CR) and CR with incomplete platelet recovery rates significantly less often than non-ETP-ALL/LBL patients (73% vs. 91%, *p* = 0.03). The median OS for patients with ETP-ALL/LBL was 20 months vs. not reached for non-ETP-ALL/LBL patients (*p* = 0.008), following different frontline chemotherapy schedules [24]. On the other hand, the GRAALL cooperative group reported that treatment intensification with allogeneic hematopoietic stem cell transplantation (allo-SCT) in ETP-ALL patients included in the GRAALL 2003 and 2005 trials reverted the early resistance to frontline chemotherapy, leading to comparable outcomes between the ETP and non-ETP subgroups [16]. The Spanish PETHEMA cooperative group could not replicate the benefit of allo-SCT intensification to improve OS in ETP-ALL patients [25]. In this series, the differences in OS and EFS between ETP-ALL and non-ETP T-ALL patients were mostly due to the greater rate of induction therapy failure, with a minimal contribution of the higher (not statistically significant) cumulative incidence of relapse observed among ETP-ALL patients vs. other T-ALL cases. In multivariate analysis, ETP-ALL was an independent adverse predictor for both OS and EFS (hazard ratios of 2.14 (95% CI: 1.29–3.53), *p* = 0.003 and 1.91 (95% CI: 1.18–3.11)), together with high white blood cell counts [25]. Lastly, the Chinese adult cohort, with 122 T-ALL patients and a frequency of 52.7% (59/112) of ETP-ALL, did not find a significantly different rate of CR between ETP and non-ETP subgroups.

### 2.5. Treatment

#### 2.5.1. Frontline Treatment

Over the years, ETP-ALL patients have been treated like other T-ALL subtypes, without major differences, at least in terms of induction chemotherapy reported in the different treatment protocols used by cooperative study groups. Most modern protocols used to treat adult T-ALL (and BCP-ALL) patients are the so-called pediatric-inspired protocols. They use pediatric-like chemotherapy regimens that rely on higher overall doses of steroids, vinca alkaloids such as vincristine, asparaginase/pegaspargase, antimetabolites, and intensive central nervous system (CNS) prophylaxis. They also include late intensifications, which basically are delayed repetitions of induction-like courses [58]. However, even following this chemotherapeutic schedule, adults with ETP-ALL respond less well to prednisone, exhibit an earlier development of drug resistance, and are more frequently MRD-positive at the end of induction therapy, resulting in higher rates of induction failure than in other ALL subtypes [16,24,25]. A higher resistance rate was also observed in pediatric ETP-ALL [14,21,22]. However, a poor early response to standard treatment might be reverted by fine response-based risk stratification and subsequent therapy intensification, as the UK children’s cooperative group showed [57,59]. The fairly good outcome of ETP-ALL patients in the UK study was partially attributed to the use of dexamethasone and pegylated asparaginase throughout the treatment. They concluded that the next step in treating T-ALL patients was not to stratify them on the basis of the ETP phenotype, but rather switch to nelarabine-based therapy followed by allo-SCT in any T-ALL patient with induction failure, persistent MRD, or relapsed disease [57,59]. Similar conclusions were reached by Sayed et al. in a study involving 103 pediatric patients [59].

##### FLAG-IDA as a Frontline Treatment

A very recent case-based report suggested that the use of FLAG-IDA chemotherapy (idarubicin, fludarabine, cytarabine, and granulocyte colony-stimulating factor) could revert the initial refractoriness of ETP-ALL patients. After the failure of standard induction treatment, three out of four ETP-ALL patients treated with FLAG-IDA achieved CR with molecular remission, and proceeded to allo-SCT [60]. In fact, the use of acute myeloid leukemia-oriented intensified chemotherapy has been effective in clearing the disease in pediatric ETP-ALL patients who were refractory to the initial induction chemotherapy [61].

#### 2.5.2. Cell-Immunotherapy: Allo-SCT

Intensification therapy in most adult treatment protocols means allo-SCT in the first complete remission (CR1) to overcome the high-risk features of ETP ALL. This has been demonstrated by the GRAALL cooperative group in a recent publication [16]. In this study, while individuals with ETP ALL had higher rates of frontline resistance, there was no statistical difference in EFS or OS when compared with non-ETP ALL cases, probably due to the increased frequency of allo-SCT in the former group (48.9% vs. 28.3%), since all patients with corticosteroid resistance, early bone marrow chemotherapy resistance, and remission induction failure were allocated to allo-SCT by design. However, the lack of complete MRD data made it impossible to prove that allo-SCT improved the outcome of MRD-positive patients [16]. Further support for the use of allo-SCT came from a multicenter study, including the University of Texas MD Anderson Cancer Center (UTMDACC), the Oregon Health and Science University (OHSU), and the National University Cancer Institute of Singapore [62]. In addition, the recommendation of allo-SCT in CR1 for all patients with immature T-ALL or mature T-ALL, and for high-risk patients with thymic T-ALL (defined as late CR, complex karyotype, or MRD positivity) was also established by the GMALL German cooperative group, with successful results [63]. A comparison of the results of the 06/99–07/03 trial (with the recommendation of allo-SCT at CR1) with those of the 05/93 trial (with no allo-SCT recommendation), an increase in the OS at 5 years from 44% to 56%, with an improvement for mature (49% vs. 30%) and immature (40% vs. 33%) T-ALL, was observed in the latter, where it was attributed to a higher rate of allo-SCT together with a substantially better conditioning regimen [64].

Ultimately, AYA (adolescent and young adult) patients deserve a separate paragraph, since this age category has been treated differently with pediatric or pediatric-inspired adult protocols with different intensification schedules and strategies. In the USA, a general approach for AYA patients with ETP-ALL is to pursue allo-SCT in CR1 if they are good candidates for transplantation and have slower clearance of MRD [15]. The European Society for Blood and Marrow Transplantation has no position regarding AYA patients but do not recommend allo-SCT for children with ETP-ALL [65,66].

#### 2.5.3. New Chemotherapy: Nelarabine

The real big caveat in the treatment of ETP-ALL patients is how to treat patients with refractory or relapsed (R/R) disease. Few options are available, and nelarabine is the only approved drug for relapsed T-ALL. The safety and efficacy of nelarabine was initially tested in pediatric ALL [19,67], and the drug eventually deserved testing in untreated patients. The large frontline study performed by the COG cooperative group assessed if nelarabine improved the outcome of intermediate and high-risk patients, if administered after induction, and if high-risk patients benefitted from nelarabine and methotrexate at two different schedules (double randomization trial) [68]. The 5-year disease-free survival (DFS) rates for patients randomly assigned to nelarabine (*n* = 323) and no nelarabine (*n* = 336) were 88.2% ± 2.4% and 82.1% ± 2.7%, respectively (*p* = 0.029). The best-performing arm was that with nelarabine and escalating doses of methotrexate (Capizzi methotrexate, vs. high-dose methotrexate), in which an additional six pegaspargase doses were given instead of four in the high-dose methotrexate arm, with a 5-year DFS of 91%. Patients who received nelarabine had significantly fewer isolated and combined CNS relapses compared with patients who did not receive nelarabine (1.3% ± 0.63% vs. 6.9% ± 1.4%, respectively; *p* = 0.0001). Toxicities, including neurotoxicity, were manageable in all four arms [68].

In adult patients, data on the safety and efficacy of nelarabine-based therapy came first from the MD Anderson Cancer center [69]; neurologic toxicity represented the main adverse event, mainly of Grade I–II. The same group assessed the potential benefit of adding this drug to hyper-CVAD frontline chemotherapy. Surprisingly, they showed no survival benefit with the addition of nelarabine to hyper-CVAD treatment. A large-dose schedule of nelarabine to avoid the potential cumulative neurotoxicity, together with the low number of patients enrolled in the study, may be the potential explanations of this result [70]. A study update showed no survival benefit with nelarabine in ETP-ALL patients, whereas the results were improved in non-ETP ALL patients [32]. Other studies on adult T-ALL have focused the use of nelarabine in post-allo-SCT relapse, reporting an overall hematological response rate of 81%, and an EFS and OS at 1 year of 70% and 90%, respectively [71]. Finally, an observational Phase 4 study provided evidence of the utility of nelarabine as a salvage therapy and bridge to allo-SCT. Here, 118 adult patients with R/R T-ALL were treated, with an overall response rate of 50% and a CR rate of 36%; 40% of the patients underwent allo-SCT with a projected OS at 2 and 5 years of 46% and 38%, respectively. The safety profile of nelarabine was acceptable, with only 8% of these cases developing Grade III–IV neurological adverse events [72].

#### 2.5.4. Target Therapy

##### Venetoclax: Monotherapy or in Combination

The potential usefulness of a *BCL2* inhibitor (venetoclax) in immature T-ALL is strongly supported by preclinical studies on cell lines and xenograft models [73,74], showing that the drug alters cell proliferation, particularly in cases with an immature phenotype. Accordingly, ETP-ALL expresses the highest levels of *BCL2* [21,26,27]. However, as the acquisition of resistance to the drug limits its use as single agent [75,76], venetoclax has been administered in combination with chemotherapy or, alternatively, with demethylating agents in R/R T-ALL, with encouraging results [77,78,79,80]. Interestingly, as the concurrent targeting of other anti-apoptotic proteins has been demonstrated to also be a rational strategy [81], combining two or even more BH3-mimetic drugs, such as BCL2L1 and BCL-W inhibitors (navitoclax, obatoclax), or MCL-1 inhibitors (bortezomib), either simultaneously or sequentially, may be an attractive approach to increase the treatment’s efficiency and to overcome resistance [82,83,84,85]. Indeed, a pilot study conducted on three patients with R/R immature T-ALL observed that the combination of venetoclax and bortezomib was an effective and well-tolerated chemotherapy-free strategy to consider, with preclinical activity confirmed by an ex vivo drug response profiling study [86].

##### Ruxolitinib

The high rate of mutations in the JAK/STAT signaling members, together with gene expression profiling, suggests the JAK/STAT pathway is constitutively activated in a subgroup of ETP-ALL cases. Thus, it makes sense to hypothesize that targeting JAK/STAT signaling would have therapeutic activity. The anti-leukemic effect of the JAK1/2 inhibitor ruxolitinib, as a single drug, has been demonstrated in ETP-ALL xenotransplants. However, in that study, the activation of JAK/STAT signaling and the efficacy of ruxolitinib were independent of mutations in the JAK/STAT pathway, raising the possibility that the therapeutic potential of the drug extends beyond the cases harboring these genomic alterations [87]. More studies need to be carried out to identify biomarkers that are predictive of response. Currently, a ruxolitinib trial is recruiting patients with R/R ETP-ALL (NCT03613428).

##### FLT3 Inhibitors

Due to the high frequency of *FLT3* mutations in adult ETP-ALL [41,88], tyrosine kinase inhibitors (TKI) already tested in *FLT3*-mutated AML [89,90] have emerged as an attractive treatment option. Data from preclinical in vitro studies have shown the effectiveness of sorafenib (a pan-TKI) in T-ALL cell lines transfected with *FLT3*-ITD and *FLT3*-*wt* constructs [41]. Interestingly, sorafenib inhibited both the wild-type and the mutated leukemic cells, suggesting an effect obtained through multiple TK-activated receptors. Although the transfection of *FLT3* expression constructs in T-ALL cell lines remains an in vitro system, the distinct sensitivity to TKIs, together with the positive experience in AML, supports the rationale for the clinical use of TKIs in ETP-ALL harboring *FLT3* mutations. A definitive evaluation of the efficacy of the *FLT3* inhibitor sorafenib in the treatment of ETP-ALL will likely be provided by ongoing clinical trials enrolling both R/R ALL and AML patients (i.e., NCT03132454, NCT01620216) (Figure 3).

## 3. Future Perspectives

In summary, we can define ETP-ALL as a subtype of immature T-ALL with a high grade of resistance to frontline treatment both in children and adults. Accurate risk stratification, together with intensified treatment schedules, was shown to revert this initial resistance frame in pediatric patients. In adult ETP-ALL patients, allo-SCT is recommended in the first CR to avoid a worse outlook due to MRD resistance or early relapse. In both adults and children, R/R ETP-ALL represents a black hole, with very few therapeutic alternatives. A breakthrough will be achieved when an in-depth genetic characterization of T-ALL allows us to distinguish distinct immunophenotypic/genomic profiles, properly classifying the unique subtypes expressing predictive biomarkers that are useful for tailored treatments, thus implementing real personalized medicine. Omics data accumulated during the last 10 years have revealed a number of potential targets. Accordingly, molecules/compounds have been developed and tested, sometimes guided by drug response profiling. Of special interest is the use of peptides that block anti-apoptotic signals (i.e., venetoclax and navitoclax). Another interesting approach related to ETP-ALL originating from an uncommitted myeloid-lymphoid progenitor and its AML-like genetics would be to use a hybrid lympho-myeloid chemotherapy frontline treatment, such as FLAG-IDA. The main purpose of these new therapeutic approaches must overcome the risk of treatment refractoriness and ameliorate the overall survival.

## Figures and Tables

**Figure 1 cancers-14-01873-f001:**
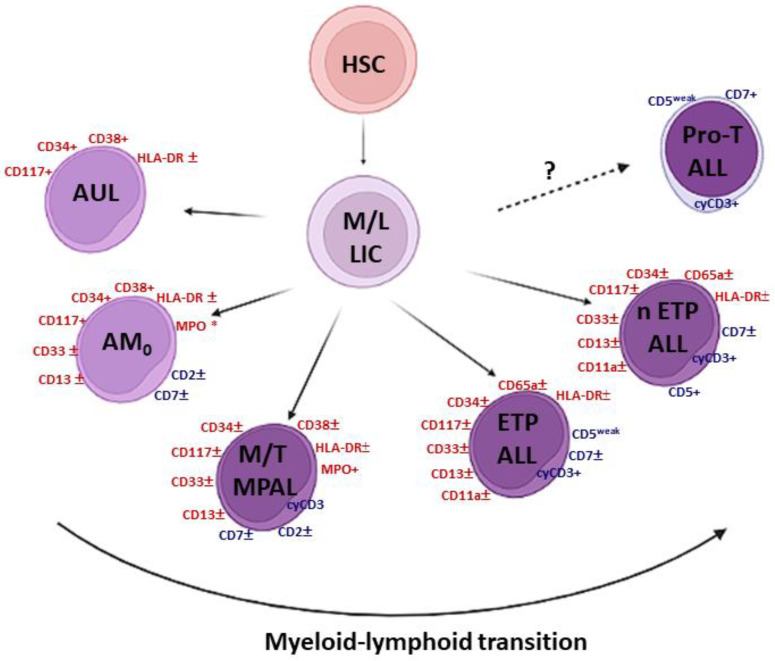
The large spectrum of immature acute leukemias. AUL: acute undifferentiated leukemia; AML_0_: acute myeloid leukemia with minimal differentiation (AML-M0 subtype according to the French–British–American classification); ±: expressed in a variable % of cases; MPO *: negative for cytochemistry but possibly positive for flow cytometry and/or immunohistochemistry; M/T MPAL: T-lymphoid/myeloid mixed phenotype acute leukemia; ETP ALL: abbreviation of early T-cell precursors of acute lymphoblastic leukemia; n ETP ALL: abbreviation of near-ETP-ALL; M/L LIC: myeloid/lymphoid-leukemic initiating cell; HSC: hematopoietic stem cell. Created with BioRender.com, accessed on 15 February 2022.

**Figure 2 cancers-14-01873-f002:**
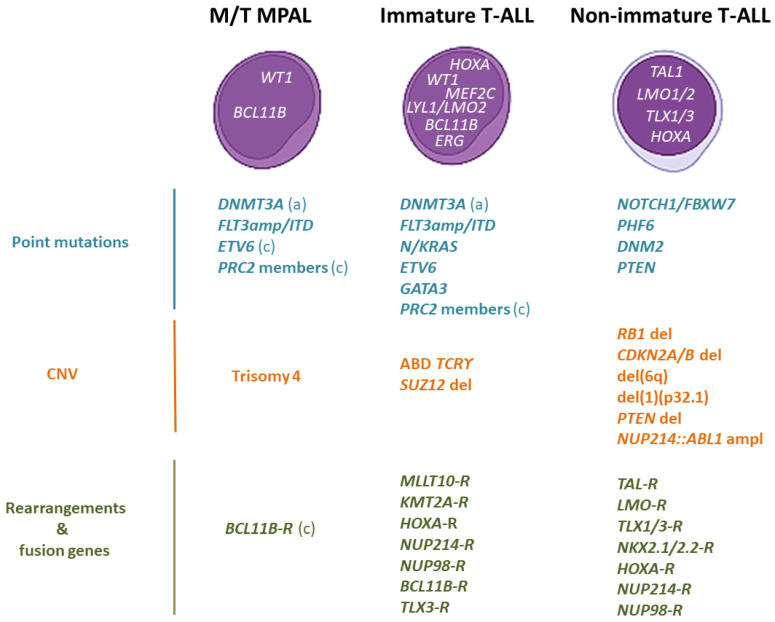
Differential genetic features across immature AL and non-immature T-ALL. Point mutations are identified in blue, CNV in brown, and rearrangements and fusions in dark green. The transcription factor delineating blast arrest differentiation are shown at the nucleus of each AL type. The immature AL subtypes are included within a square. (a): adult; (c) childhood; -R: promiscuous partner. Made with the help of BioRender.com, accessed on 15 February 2022.

**Figure 3 cancers-14-01873-f003:**
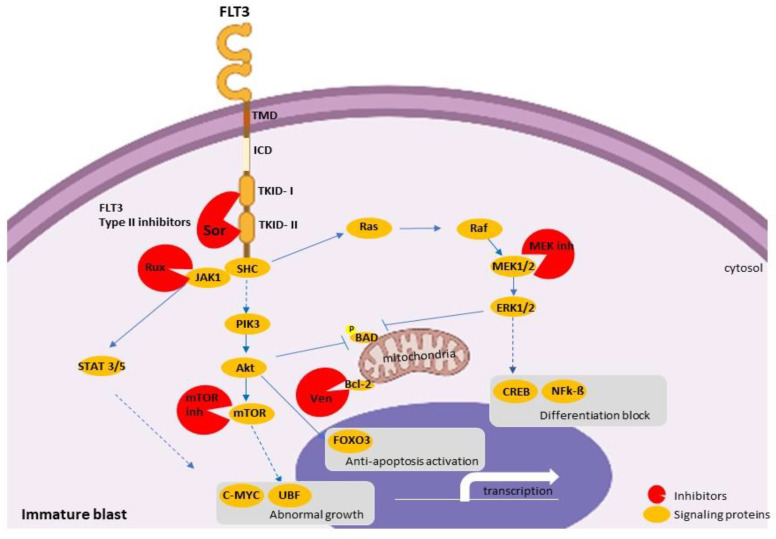
Schematic representation of the possible inhibitory points in the FLT3 signaling pathway. Type II FLT3 inhibitors such as sorafenib act in the inactivated FLT3 receptor (represented as an FLT3 monomer). Active FLT3 transduces signaling through (1) the Ras/Raf/MEK/ERK pathway, which, in turn, can be inhibited by pan- or specific MEK inhibitors; (2) the PIK3/Akt/mTOR pathway, which is also inactivated by specific mTOR inhibitors; and (3) JAK/STAT signaling, which might also be inhibited by the specific JAK1/2 inhibitor ruxolitinib. In addition, venetoclax can block the anti-apoptotic signaling effects of Bcl-2. MEK, mTOR, JAK, and Bcl-2 inhibitors can also be used in any T-ALL cases showing activation of the target proteins. TMD: transmembrane domain; ICD: intracellular domain; TKID-I and II: tyrosine kinase domain I and II; Sor: sorafenib; Rux: ruxolitinib; Ven: venetoclax. Created with BioRender.com, accessed on 16 February 2022.

**Table 1 cancers-14-01873-t001:** Immunophenotypic features of immature leukemias.

	CD3 *	MPO	CD19	Stem Cells	Other Myeloids	Other Lymphoids
AUL	−	−	−	+	−	−
AML with minimal differentiation	−	−	−	CD117+, CD38+	HLA-DR± CD33±, CD13±	CD7±, CD2±
T/M, MPAL, NOS	+	+	-	+/−	+/−	CD7±, CD2±

AUL: acute undifferentiated leukemia; * cytoplasmic and/or surface; AML: acute myeloid leukemia; T/M, MPAL, NOS: mixed phenotype acute T-myeloid, not otherwise specified; +: positive; −: negative; +/−: positive or negative. Other myeloids might also include monocytic markers, i.e., CD11c, CD14, CD36, CD64, or lysozyme; other lymphoids might also include TdT.

**Table 2 cancers-14-01873-t002:** Immunophenotypic features of T-ALL at different stages of differentiation.

	cCD3	sCD3	CD7	CD1a	TdT	CD2	CD5	CD4/CD8	Stem Cell/Myeloid
ETP-ALL	+	−	+	−	±	−	±	−/−	+/− or −/+ or +/+
Near-ETP-ALL	+	−	+	−	±	−	+	−/−	+/− or −/+ or +/+
Pro-T ALL	+	−	+	−	± or +	−	−	−/−	−
Pre-T	+	±	+	−	± or +	+	+	−/− or +/+	−
Cortical	+	±	+	+	±	+	+	±/±	−
Mature	+	+	+	−	± or −	+	+	+/− or −/+	−

ETP-ALL: early T-cell precursor acute lymphoblastic leukemia; c: cytoplasmic; s: surface; +: positive; −: negative; ± and +/−: positive or negative; CD5±: negative or expressed by less than 75% of blasts; stem cell: CD34 or CD117 (at least 25% of blasts); myeloid: CD13, CD33, HLA-DR, CD11b, or CD65 (at least 25% of blasts).

**Table 4 cancers-14-01873-t004:** Genomic rearrangements underlying *BCL11B* activation.

Molecular Mechanism [9,53]	Type of Alteration [9,53]
**Chromosomal rearrangement**	**Fusion transcript**
t(2;14)(22.3;q32.2)	*ZEB2::BCL11B*
**Chromosomal rearrangement**	**Enhancer hijacking**
t(3;14)(p24.3;q32.2)	*SATB1::BCL11B*
t(6;14)(q25.3;q32.2) ins(6;14)(q25.3;q32q32)	*ARID1::BCL11B*
t(7;14)(q22.1;q32.2)	*CDK6::BCL11B*
t(8;14)(q24.2;q32.2)	*CCDC26::BCL11B*
t(12;14)(p13.2;q32.2)	*ETV6::BCL11B*
t(14;21)(q32.2;q22.1)	*RUNX1::BCL11B*
**Focal amplification**	**New enhancer**
14q32 non-coding sequence	BETA_*BCL11B*

BETA: *BCL11B* enhancer tandem amplification.

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
