# Peer review of "Early T-Cell Precursor ALL and Beyond: Immature and Ambiguous Lineage T-ALL Subsets"

_cancers, 2022, doi:10.3390/cancers14081873_

Round 1
Reviewer 1 Report
The manuscript titled “Early T-cell precursor ALL and beyond: immature and ambiguous lineage T-ALL subsets” describes ETP-ALL as a subtype of immature T-ALL with a high grade of resistance to frontline treatment in both children and adults. The authors summarized new genetic information that can be used for accurate diagnosis of this high-risk mortality disease, and prospect novel personalized medicine and more efficient treatment are eagerly needed to improve the overall survival rate.
Overall, the manuscript is well written for the most part. I only have some marks listed below that the authors may want to consider.
Major Concerns and Comments:
- Line 72: Provide reference for “roughly 40% of cases”.
- Lines 232-237: Is LYL1 involved in one of the highly expressed genes in immature T-ALL? I’d suggest the authors make it clear whether LYL1 is involved in the four genes listed on line 234 or not. Additionally, is there any further information between “PSCD4 and MEF2C”, “HHEX and MEF2C”, and “FAM46A and MEF2C”?
- Table 3: Please add the reference to each interaction in the table for easier tracking instead of only listing the references at the bottom of the table.
- Please add references to Table 4 for easier tracking.
- Line 371: The “95% CI” is missing.
Minor Concerns and Comments:
- Figure 1: Use “Pro-T ALL” in the image instead of “Pro-T”.
- Line 80 “near ETP ALL”, line 84 “near-ETP ALL”. Please check throughout the whole manuscript and homogenous the format.
- Please use italic p as it refers to a p-value. For example, line 224, line 226, line 331, line 333, lines 343-344, line 357, line 451, line 457, and so on. Check throughout the whole manuscript.
- Line 232 “in vitro” is not italic; line 247 “in vitro” is italic. Please check throughout the manuscript and homogeneous the format.
- Line 173 “Table 2” is not bold, line 271 “Table 3” is bold.
- Line 356: The “CR” should be defined here. Complete response or other.
- Line 370: Just use “CI” instead of “confidence interval” as everywhere else.
- Line 340, the authors described “More recent data” for the reference [56] was published in 2014; while on line 412, the authors described “a recent publication” for the reference [15] was published in 2017. Please rephrase the wording.
- Line 458: There is an extra “-“ between “arms” and “[67]”.
- Line 460: Please update the citation “Jain P, leukemia 2014”.
- Line 525 Figure 3: Provide the software or online tools that support the generating of signaling pathway images.
Reviewer 2 Report
As the authors indicate, this review focuses on genetic events associated with immature T-ALL and outlines their relationship with treatment and outcome. In authors´ words, the goal is to offer a basis to use the genetic information for new diagnostic algorithms, in order to better stratify patients and improve their management with more efficient and personalized therapeutic options.
The authors provide a thorough compendium of updated data, clearly structured and easy to follow. The text is clear and complete, most figures and tables are really useful and the bibliography is very updated and well selected.
I have some minor concerns:
- Page 10, lines 335-336: it would be good to indicate how many of those 100 and 91 were actually ETP patients.
- Page 10, line 327: the section “Outcome” referred to ETP does not include any information from the TARGET cohort (doi:10.1038/ng.3909), with 189 patients classified into ETP (n=19), near-ETP (n=24) and non-ETP (n=146). Why did the authors choose not to mention this study?
- Figure 2 legend: it says “The immature AL subtypes are included within a square.” What does it refer to?
In addition, I have some suggestions that might help to improve the clarity of the article:
- Table 1: nomenclature should stick to what has been previously stated in page 2 and Figure 1: T/M MPAL for T/Myeloid mixed phenotype acute leukemias. Moreover, it should be consistent throughout the text (in page 6, line 203, it is again stated as T/Myeloid MPAL.
- In Table 1 legend, additional myeloid and lymphoid markers are stated, why not to include them in the actual table?
- Page 6, line 238: this paragraph is somewhat confusing. It would be of help if the authors tried and re-write it.
- Figure 2: nomenclature: “Immature vs Typical T-ALL” is somewhat confusing. I understand that by Immature the authors refer to ETP and near ETP T-ALL, but maybe it´s better to say non-immature, not-ETP or post-ETP (or even Pre-T, Cortical & Mature T-ALL) instead of “Typical”.
- Regarding section “2.5. Treatment”, I think that it could be useful to include a table gathering all the results referring clinical outcomes.
Regarding format and language, I have only detected very few mistakes:
- Some typos:
- page 2, line 50, a hyphen.
- Page 4, line 130, the symbol for gamma cannot be seen.
- English mistakes:
- page 4, lines 111-112 (… with “immature T-ALL” we refer to cases with slightly different patterns of antigenic expression, which however, is not well known to reflect true distinct clinical and prognostic subsets)
- page 10, line 333 (“significative”)
